# Huanglongbing as a Persistent Threat to Citriculture in Latin America

**DOI:** 10.3390/biology14040335

**Published:** 2025-03-25

**Authors:** Jael Arely Cervantes-Santos, Hernán Villar-Luna, Ana Marlenne Bojórquez-Orozco, José Ernesto Díaz-Navarro, Ángela Paulina Arce-Leal, María Elena Santos-Cervantes, Manuel Gonzalo Claros, Jesús Méndez-Lozano, Edgar Antonio Rodríguez-Negrete, Norma Elena Leyva-López

**Affiliations:** 1Departamento de Biotecnología Agrícola, CIIDIR Unidad Sinaloa, Instituto Politécnico Nacional, Guasave 81101, Mexico; jcervantess1400@egresado.ipn.mx (J.A.C.-S.); hvillarl2200@alumno.ipn.mx (H.V.-L.); abojorquezo1800@alumno.ipn.mx (A.M.B.-O.); jdiazn2300@alumno.ipn.mx (J.E.D.-N.); angela_paulina22@hotmail.com (Á.P.A.-L.); msantos@ipn.mx (M.E.S.-C.); jmendezl@ipn.mx (J.M.-L.); erodriguezn@ipn.mx (E.A.R.-N.); 2Institute for Mediterranean and Subtropical Horticulture “La Mayora” (IHSM-UMA-CSIC), 29010 Malaga, Spain; claros@uma.es; 3Department of Molecular Biology and Biochemistry, Universidad de Málaga, 29010 Malaga, Spain; 4CIBER de Enfermedades Raras (CIBERER) U741, 29071 Malaga, Spain; 5Institute of Biomedical Research in Málaga (IBIMA), IBAMA-RARE, 29010 Malaga, Spain

**Keywords:** Huanglongbing (HLB), *Candidatus* Liberibacter genus, Latin America, HLB epidemiology, HLB management methods

## Abstract

Huanglongbing, also known as citrus greening, is a bacterial disease transmitted by insects that has severely impacted citrus production globally. This review delves into the repercussions of HLB in Latin America, analyzing its dissemination, genetic evolution, and potential management strategies. The goal is to provide Latin American citrus growers with the tools they need to fight the devastating effects of this disease and ensure the sustainability of the citrus industry in the region. Given the transnational nature of this pest, it is imperative to strengthen collaboration between countries and invest in research to deal with this persistent threat to citrus farming in Latin America.

## 1. Introduction

Citrus fruits are among the most traded globally due to their extensive cultivation and high consumption. Oranges are the most traded citrus fruits, followed by tangerines, lemons, and grapefruits [1]. The largest citrus producers worldwide are China, Brazil, India, the United States, and Mexico. Nowadays, citriculture faces whopping challenges in dealing with diseases that cause a huge economic loss to the industry, such as Huanglongbing (HLB), which has become one of the most devastating and unprecedented diseases due to the interaction between the pathogen, plant, and vector in citrus crops across the world [2,3].

HLB has been associated with three species of the *Candidatus* Liberibacter genus: *Candidatus* Liberibacter asiaticus (*C*Las), *Candidatus* Liberibacter africanus (*C*Laf,) and *Candidatus* Liberibacter americanus (*C*Lam). This genus falls under the Gram-negative and α-proteobacteria, belonging to the family *Rhizobiaceae* [2]. Symptoms related to HLB have historically been mistaken for those caused by nutritional deficiencies; nevertheless, early symptoms appear as yellow shoots, blotchy mottle leaves, aborted seeds, and high levels of starch in the leaves [4]. The transmission of *Candidatus* Liberibacter associated with HLB is mainly facilitated by two psyllid vectors: *Trioza erytreae* and *Diaphorina citri* [5].

The HLB disease was first observed in Southeast Asia in 1870 [6], currently the disease has spread to around 67 countries in Africa, Asia, Oceania, and America (South, North, and Central America, and the Caribbean) [7]. Since the arrival of HLB in Latin America in 2004 in the Brazilian state of São Paulo, the disease has spread to 26 counties. The impact of HLB in Latin America has generated large economic losses. In Brazil, the disease is responsible for the eradication of more than 55 million HLB-infected trees. Additionally, in Mexico, HLB reduced the yield of infected trees by up to 50%. Considering that Brazil and Mexico are two of the main citrus-producing countries, the reduction in production had a direct impact on employment, processing, and the distribution of citrus fruits [8,9]. To date, there are no reports of citrus cultivars with resistance or tolerance to the disease [5,10]. And disease management is focused on preventing its spread by chemical and cultural management [11]. In addition, the pursuit of resistance or tolerance continues to be a crucial approach for alleviating or managing the effects of the disease. Therefore, recently, enormous efforts to generate tolerant citrus cultivars using modern “omics” approaches and genetic transformation/edition have been implemented [12,13,14]. Despite the substantial knowledge generated on the HLB epidemiology, *Candidatus* Liberibacter genetics, vector control, molecular citrus–*Candidatus* Liberibacter interactions, and integrated management tools, most of this research has been conducted in North America and China, while it remains limited in Latin American citrus-producing countries.

## 2. Epidemiology

To date, three *Candidatus* Liberibacter species associated with HLB disease have been identified, with some different characteristics, but causing the same symptoms: *Candidatus* Liberibacter africanus (*C*Laf), *Candidatus* Liberibacter *asiaticus* (*C*Las), and *Candidatus* Liberibacter americanus (*C*Lam) [2]. *C*Las is the only heat-tolerant species (≤35 °C) and is widely distributed throughout the world [15] (Table 1).

In addition to the *Candidatus* Liberibacter genus, the characteristic symptoms of HLB disease have also been associated with the *Candidatus* Phytoplasma genus in Brazil, China, Mexico, and Chile (Table 2).

The natural transmission of *C*Las and *C*Lam is mediated by the insect vector *Diaphorina citri* kuwayama, but experimental transmission can also be achieved by grafting infected budsticks or by the parasitic plant *Cuscuta* sp. [27]. One important way to spread the bacteria over long distances is through the marketing of contaminated vegetative material, which is one reason why some countries are affected by the disease [2]. Mexico has implemented NOM-079-FITO-2002, which guarantees that the plants used are free of *C*Las and other pathogens. Brazil has implemented IN-SDA-MAPA 10, which established producing propagation material in protected environments [28]. Argentina has implemented SENASA Resolution No. 930, which established that the production of citrus plants must be under conditions of isolation, within a nursery covered with anti-aphid mesh to prevent the entry of the insect vector [29]. The bacterium *C*Las is thought to be transmitted in a persistent-propagative manner by *Diaphorina citri*, observing an increase in titers of the bacterium in nymphs, but they can also be transmitted in late stages [30,31,32]. The infected citrus plants are more attractive to *D. citri* because they present a high concentration of volatile organic compounds, such as MeSA and β–caryophyllene [33]. *C*Las can be acquired after 15 to 30 min of feeding on an infected citrus fruit, but an efficient acquisition requires longer feeding periods, and once the bacterium is acquired by the insect, it can be detected throughout its life because *C*Las is found in several organs such as the midgut, hemocoele, fat body, ovaries, and salivary glands [34,35]. *D. citri* tends to disperse from plant to plant over distances of 30 to 100 m [36]. Although the distribution of the bacterium is not homogeneous in the plant, it has been reported that there is a higher *C*Las titer in the phloem sieve of asymptomatic young flushes, while in mature leaves with symptoms, bacterial cells are found in a non-viable state. The ability of adults to feed on mature leaves compared to nymphs, which concentrate their feeding activities on young leaves, may reduce the probability of adults ingesting viable bacterial cells, therefore contributing to a lower rate of *C*Las acquisition than nymphs [2,37,38]. The average latency period of *C*Las in *D. citri* is between 16 and 18 days at 25 °C. During the latency period, psyllids are unable to transmit *C*Las, thereby providing an opportunity for vector control measures to prevent the spread of pathogens [39] (Figure 1).

In America, the first case of the Asiatic form of HLB was in South America, in the Araraquara municipality, a central region of São Paulo State, Brazil, in March 2004 [40]. Thereafter, the disease spread to all citrus regions in São Paulo and later to other neighboring states such as Minas Gerais in 2005, Paraná in 2007, Mato Grosso do Sul in 2019, Bahia in 2020, and Santa Catarina in 2022, as well as Argentina in 2012, Paraguay in 2013, Colombia in 2015, Venezuela in 2018, and Uruguay in 2022 [7,11]. In Mexico, the first report of the Asiatic form of HLB was in the Yucatán Peninsula in 2009 [41]. In 2010, new infections were reported in states such as Campeche, Colima, Sinaloa, and Michoacán. Currently, the disease has been detected in 351 municipalities in 25 states of Mexico, of which 292 are citrus-producing municipalities [42]. Based on the occurrence and intensity of the disease, citrus-producing areas were categorized into two regions: the Pacific region (high intensity) and the Yucatan Peninsula (low intensity) [43]. The HLB disease has also spread in several Caribbean and Central American countries, including Cuba and Puerto Rico in 2007, the Dominican Republic in 2008, Belize and Jamaica in 2009, Guatemala, Honduras, Nicaragua, and the US Virgin Islands in 2010, Costa Rica in 2011, Dominica and Guadeloupe in 2012, El Salvador and Martinique in 2013, Barbados in 2014, Panama in 2016, Trinidad y Tobago in 2017, and French Guiana in 2021 [7]. Although the disease is present in most of the Caribbean and Central American countries, it is only widespread in Jamaica, Cuba, and Puerto Rico [7]. The American form of HLB was reported only in South America, in São Paulo State, Brazil, in March 2004 (Figure 2) [15].

Therefore, spatial–temporal studies of HLB have been conducted with the objective of determining the dispersion pattern of the disease. They found that there is a similar pattern of dispersion in the United States and Brazil. The spread of the disease in Brazil was highly explosive due to its agroclimatic conditions, starting from its initial detection in 2004 and reaching a dispersion radius of 100 km within the first year. In Mexico, the dispersion gradient reached 100 km in a span of 1.5 years. The arrival of HLB was probably due to the movement of the insect vector from countries with the presence of the disease, such as Belize, Cuba, and the Dominican Republic. This could probably have been caused by the winds in that area. On the other hand, in the Pacific area, it was due to the displacement of infected plant material [44]. Citriculture is of great economic importance for Mexico, with an approximate value of MXN 56 billion, ranking second in the production of limes and lemons and fourth in oranges worldwide [1,9]. The most critical year for Mexican citrus after the arrival of HLB was 2014, reducing its production by 30 percent. The state of Colima contributed 48 percent of the national production of Mexican lime; by 2014 it reduced significantly to 17 percent [45]. In Mexico, *D. citri* was detected in 2001 and 2002 in the states of Tamaulipas, Campeche, and Quintana Roo; only 6 years later, it was dispersed in almost all citrus-growing areas of the country [46]. There are few studies regarding the disease spreading in Latin America, especially in Mexico. Some of these studies focus on the insect vector, as demonstrated by López-Collado et al. [47], who concluded that the most favorable regions for the insect vector are located in the Gulf of Mexico region, the Yucatan Peninsula, and scattered areas of the Pacific coastal states. The variables that most affect the distribution are the minimum temperatures of the coldest months, the range, and the seasonality of the temperature. Mexico has a large neotropical zone, and as a result of climate change, the temperature is expected to increase in the near future. Future projections coincide with an increase in the risk of HLB in new areas that are not currently dedicated to citrus farming, which are contiguous to the areas that already present the disease and from which the insect vector could spread the disease to production areas [48]. It is suggested that the rapid spread of the insect vector is due to its ability to adapt. *D. citri* displays phenotypic plasticity in response to diverse temperature conditions and its capacity to utilize diverse hosts, thereby altering its body size and morphology [49]. Given that the Americas have diverse geographic regions, it is likely that additional evolutionary factors were involved. There is the genetic diversity of *D. citri* among six localities in the states of Jalisco and Colima in Mexico, and low genetic diversity and intermediate differentiation were found among all populations. These populations were clustered into two genetic groups. Both genetic groups were present in the state of Jalisco; however, the first group was confined to El Arenal, Jalisco, while the second group encompassed populations from Jalisco and Colima. On the other hand, Yucatán populations shared variation between the two groups [49].

In Brazil, the world’s leading producer of oranges, HLB disease was first reported in March 2004 in Araraquara, São Paulo state. The first detection of the genus *Liberibacter* was *C*Las, but in April 2004, a new species of *Liberibacter* called *Candidatus* Liberibacter americanus (*C*Lam) was discovered, being the most widely distributed pathogen in that region [15,40]. However, about four years later, it was almost completely replaced by *C*Las. Since 2004, research on HLB became a priority in Brazil. Before that year, *D. citri* was already present, but was not managed because without HLB-associated bacteria, it was a non-relevant citrus pest [15,50]. In Brazil, the implemented system to stop the HLB spread consisted of planting certified healthy material, removing infected trees, and controlling the insect vector. The Brazilian strategy of sustaining citrus production and the competitiveness of Brazilian citrus farming have been considered successful in countries where HLB disease has reached epidemic levels. The incidence of the disease between 2008 and 2012 almost doubled. However, between 2015 and 2019, a slight increase of 17–19% was observed. Efforts to control HLB in Brazil have managed to almost double the productivity of sweet oranges in the 2019/2020 season (1051 boxes/ha) compared to the 2004/2005 season (616 boxes/ha) [51]. Studies have been conducted on the incidence of *C*Las-positive *D. citri* in the citrus region of Brazil, showing that the averages of positive psyllids were highest in the southwestern region of São Paulo at 74%, continuously decreasing from the center, north, and state border between São Paulo/Minas Gerais (67%, 56%, and 33%). An increase in the psyllid population was observed in the summer and autumn in the central and northern regions of the Brazilian citrus belt, contrary to what was observed in Colima, Mexico, where the averages of positive psyllids were higher during the winter and spring months. Temperature influences the titer of the bacterium and that can lead to a lower acquisition and incidence of the bacterium in *D. citri* [52,53,54,55,56].

On the other hand, Argentina ranks fourth in the production of limes and lemons worldwide [1]. In June 2012, HLB was confirmed in tangerines and ‘Rangpur’ lime backyard trees in the Northern part of the Misiones Province, across the border from Brazil and Paraguay [57]. The arrival of HLB in this country could be due to the introduction of infected plant material from neighboring countries where the disease is present. However, the disease was present with a low incidence and was managed only by eliminating infected plants [58]. In 2017, *C*Las was detected for the first time in *D. citri* in the city of Ituzaingó in the Province of Corrientes. The sequenced PCR products confirmed the first detection of *C*Las in Argentina. Due to these results, HLB control measures such as the biological and chemical control of the vector insect and the removal of infected trees at the early stages of infection were implemented [59].

## 3. Genetic Diversity

The genetic diversity of *C*Las comes from a variety of sources, including the geographical location, citrus cultivars, and evolution [60]. Studying the genetic diversity of *C*Las is important for disease management and understanding the spread of HLB [61]. Currently, the research about the characterization of *C*Las strains is based on molecular methods due to the inability to culture it in vitro. The first molecular regions used for the genetic characterization of *C*Las were the 16S rRNA gene, 16S/23S rRNA spacer regions, the rplKAJL-rpoBC, nusG-rplK operon sequence, and the *omp* gene region. These *loci* have been helpful in detecting and differentiating *C*Las species, but they are very limited in identifying variations in the strains and, therefore, are not suitable for genetic differentiation studies [62,63,64,65]. Recently, *C*Las genomic studies have provided genome sequences that have facilitated the development of molecular markers to study its population diversity [66,67]. Chen et al. [68] characterized the genetic variation of *C*Las strains from Guangdong, China, and Florida, USA, where the disease was first observed around 100 and 20 years ago, respectively, using a hypervariable genomic *locus*, CLIBASIA_01645, containing tandem repeat numbers (TRNs) from the repeat unit AGACACA. The authors found that in Guangdong and Florida, populations predominated *C*Las strains with TRNs 7 and 5, respectively. Subsequent studies conducted in China, Florida, Brazil, and Mexico showed evidence that the populations in these countries are different because of the varying distribution of TRN frequencies using the repeat units TACAGAA and AGACACA (Figure 3) [69,70,71]. The presence of predominant *C*Las-respective TRN types in each country suggests selection pressure for biotic and abiotic factors that are yet unknown in the bacterial populations. The more genetic diversity has been reported in the China population because the HLB disease has been there longer [68,69,72]. In Florida, two *C*Las genotypes have been reported. One genotype is more prevalent than the other, with the most dominant genotype located in the southern part of Florida, showing a predominance of five and nine TRN frequencies using the repeat units AGACACA and TACAGAA, respectively. The second dominant *C*Las genotype was found in Polk County, situated in Central Florida, as well as in the neighboring counties of Marion and Alachua to the north. The more prevalent *C*Las genotype found in Florida was also detected in the Caribbean and Central American countries, while the same two genotypes from Florida were also identified in southern Mexico [70]. However, a more extensive study conducted in the main citrus-growing areas of Mexico revealed a greater diversity of TRNs than in Florida (Figure 3) [69,71]. This diversity may be attributed to the inclusion of regions containing both sweet (oranges, grapefruits, and tangerines) and sour (limes and lemons) citrus samples, subjected to diverse environmental conditions. Previous research has suggested that the coexistence of different TRN genotypes may be associated with bacterial adaptation to the environment [68]. Two genogroups were identified in Mexico by a recent study based on next-generation sequencing and machine learning analysis. Genogroup 1 comprises *C*Las strains from central and northern Mexican states and one from California, USA. Additionally, genogroup 2 comprises *C*Las strains from southern Mexican states, the Yucatán Peninsula, and one from Florida, USA. These results support the hypothesis that *C*Las may have been introduced to Mexico through two separate events [73].

There are few studies that revealed the movement pattern of *C*Las worldwide. One of the widest studies to identify *C*Las strains from American countries (USA, Florida, and Brazil), and Asian countries (China, Cambodia, Vietnam, Thailand, Taiwan, Japan, and India) was performed by Islam et al. [69] using tandem repeat-based analysis. They suggested three founder haplotypes because the *C*Las strains were clustered into three groups. All the Brazilian (South America) *C*Las strains formed part of the same group, including the East and Southeast Asian strains, together with a few strains from North America (Florida). However, most Florida strains were grouped separately, and the Indian strains formed part of another group genetically distinct. Due to the similar genetics of east–southeast Asian and Brazilian strains, the authors suggest that *C*Las was probably introduced to Brazil from East–Southeast Asia [60,72]. A subsequent study demonstrated the population homogeneity of Brazilian *C*Las strains [74]. Similarly, double-locus (DL) analyses based on TRN and SNP (Single Nucleotide Polymorphism) in the prophage *loci* CLIBASIA_01645 and CLIBASIA_05610 has been used to determine the *C*Las diversity in Brazil, the USA, and China [72]. Their results show three DL genotypes based on the geographical origin. DL genotype 1 was significantly predominant in Chinese strains (97%), DL genotype 2 included all Brazilian strains, although DL genotype 1 was found later with a low incidence (1.21%) [74], and DL genotype 3 comprised 93% of the USA (Florida) strains. Since the California strains belonged to Asiatic DL genotype 1, these strains were likely introduced into California from China. However, when and how the *C*Las strains causing the HLB disease were introduced into the Western Hemisphere is yet unknown. The HLB disease has spread all over South, North, and Central America and the Caribbean countries in a relatively short period of time, and no identical haplotypes of *C*Las were found on the Asian and American continents or even between the countries within the same continent. These results suggest the possibility of the contemporary migration of the *C*Las strains among the countries through the movement of infected propagating material or by the migration of the vector *D. citri.* A rapid mutation and selection could lead to population deviation from their original sources (Figure 3) [69].

## 4. Omics Studies to Decipher Citrus–*Candidatus* Liberibacter Interaction

To date, HLB disease management conventionally resides in psyllid vector chemical control and crop protection practices, including the production of pathogen-free plants, the removal of infected trees, and nutritional management [75]. Consequently, the ultimate long-term HLB control strategy relies on developing new citrus cultivars displaying HLB resistance. However, conventional breeding is challenging because, to date, no resistance genes have been described in citrus species. To overcome this limitation, modern biotechnological approaches, including transgenesis and genome editing through CRISPR, represent promising alternatives; however, the application of these technologies requires a basic knowledge of the multilayered pathways involved in both the pathogenic mechanisms of invading bacteria and citrus host defensive responses to pathogen infection.

Recent advancements in bioinformatics and high-throughput technologies have resulted in the emergence of Omics sciences, facilitating the comprehensive examination of living organisms at various molecular levels. Omics sciences are divided into four major branches: genomics, transcriptomics, proteomics, and metabolomics, focused to generate a comprehensive analysis targeting the genome, transcriptome, proteome, and metabolome, respectively (Figure 4). For citrus–*Candidatus* Liberibacter pathosystems, these powerful Omics technologies are leading to the application of molecular knowledge for the generation of cost-effective tools and environmentally friendly strategies for crop improvement, ensuring citrus sustainable agriculture.

### 4.1. Genomics

For citrus models, the genome platform is critical to identify genetic polymorphisms and potential key genes, particularly those that control interesting and specific traits, and also to develop molecular markers used for assisted breeding [76]. Additionally, genomics has become invaluable in deciphering the genomes of both host plants and pathogens, enabling the identification of genes related to disease resistance [77]. These goals have been achieved using cost-effective next-generation short-read sequencing technologies, such as Illumina-based sequencing, complemented by more powerful long-read sequencing technologies, such as PacBio, which excel at obtaining reads from repetitive elements.

#### 4.1.1. Citrus Genomics

The genus *Citrus* belongs to the *Rutaceae* family, and it has been proposed that the center of the origin of citrus species is located in Southeast Asia [78]. To date, at least 7 species and 58 citrus accessions have been sequenced and are available at the Citrus Genome Database (https://citrusgenomedb.org accessed on 15 November 2023), with *C. clementina* genome being the first reference genome for the Citrus genus.

In citrus, asexual reproduction and propagation causes low genetic diversity, making citrus prone to suffering diverse diseases, including HLB, where most commercial citrus cultivars are susceptible, displaying variable symptom degrees, with generally sweet orange and grapefruit being susceptible, and some lemons and limes semi-tolerant [2,79,80,81]. Therefore, it is imperative to explore wild germplasms and citrus-relative species to improve HLB tolerance in citrus cultivars through structural, comparative, and functional genomics. Important public resources, including Expressed Sequence Tag (EST) databases, genetic linkage maps, and other tools for functional genomics, have been generated by the sequencing of whole citrus genomes thanks to the International Citrus Genome Consortium [82].

An intergeneric F1 population of HLB-susceptible *C. sinensis* and a highly tolerant close citrus relative (*Poncirus trifoliata*) was genotyped by genotyping-by-sequencing, and high-density SNP-based genetic maps were constructed, allowing the detection of QTLs associated with HLB tolerance [83].

Recently, genome-wide association mapping and the analysis of allele-specific expression between more than 300 citrus accessions, including susceptible, tolerant, and resistant varieties, allowed the construction of the citrus pan-genome and the identification of determinants for HLB pathogenicity, including factors associated with the host immune response, reactive oxygen species (ROS) production, and antioxidants [84].

#### 4.1.2. *Candidatus* Liberibacter Genomics

For *Candidatus* Liberibacter, the identification of potential molecular targets has been particularly challenging due to the uncultivable nature of the bacteria. Thus, the knowledge pertaining to its physiology has been derived from in silico predictions resulting from the information encoded in its genome [85]. Nowadays, genome sequencing has been obtained from six *Liberibacter* species, including *C*Las, *C*Laf, *C*Lam, *Candidatus* Liberibacter solanacearum (*C*Lso), *Liberibacter crescens* (Lcr), and *Candidatus* Liberibacter europaeus (*C*Leu). On the American continent, all of them have been detected except for *C*Laf and *C*Leu [86], with *C*Las being the most commonly found and *C*Lam only reported in Brazil [15]. Until now, there have been 23 American whole genomes of *C*Las, 18 from the USA, one from Colombia, one from Brazil, and three from Mexico; four *C*Lso (from the USA), two *C*Lam (from Brazil), and two Lcr (from Puerto Rico) are available in the GenBank database (version 249) (https://www.ncbi.nlm.nih.gov/genbank/release/249/ accessed on 15 January 2025) (Table 3).

*Liberibacter* genomes range from 1.17 to 1.52 Mb in size, with a low GC content of 33.8% average. A phylogenetic analysis of the *Liberibacter* species and eight related Rhizobiales species suggested the first evolution step from a common ancestor into nonpathogenic *L. crescens* (Lcr), followed by a second evolutive step to pathogenic *Ca.* Liberibacter spp. [87]. Since tremendous efforts to culture *Ca.* Liberibacter spp. associated with HLB in artificial media have failed, genomic tools could offer new options to achieve such a goal. A comparative analysis between the Lcr genome and other *Liberibacter* species showed that Lcr encodes more genes for thiamine and other essential amino acids production, which explains why Lcr is culturable while other *Liberibacter* species are not [88]. Additionally, it has been found that there are missed genes in the *C*Las genome that avoid its growth in artificial media, which may not be easily solved by an adjustment of the media composition. Whereas the glyoxalase pathway, which prevents both eukaryotic and prokaryotic cells from proteome glycation and methylglyoxal-induced carbonyl stress, is functional in culturable and nonpathogenic Lcr, and those genes are not functional in all unculturable and pathogenic *Liberibacter* species [89]. Therefore, it has been suggested that the addition of a specific methylglyoxal compound to the culture medium or the transferring of the glyoxalase pathway-related *glo A* gene from Lcr to *C*Las could make *C*Las culturable in axenic media [90,91]. In addition, the very-long-chain fatty acid (VLCFA)-modified lipid A homologous genes *LpxXL* and *AcpXL* are present in Lcr but absent in other *Liberibacter* species. Interestingly, the mutation of the Lcr *LpxXL* gene is lethal, suggesting that VLCFA-modified lipid A could be necessary for *Liberibacter* growth in axenic media [92].

Furthermore, the availability of *Liberibacter* genomes has accelerated the understanding of pathways related with bacterial growth and pathogenicity. The reduced *C*Las genome size suggests that the pathogen heavily depends on the host metabolic machinery [93]. A reconstructive metabolic model analysis of six *C*Las strains indicated that the pyrimidine/purine metabolism, fatty acid metabolism, and gluconeogenesis are the most common essential genes for bacterial survival [94]. Metabolic studies have suggested that broad changes in sugar concentrations occur in the leaves and fruit of *C*Las-infected plants [95]. *C*Las is able to metabolize xylulose, glucose, and fructose, but not galactose, mannose, rhamnose, and cellulose. Due to the very low concentration of glucose and fructose in the phloem sap, *C*Las requires the initiation of a shift in the host sugar distribution during infection [66]. Comparative genomic studies have revealed the presence of a large number of transporter proteins in *C*Las, including 14 ABC transporter-related proteins, which might play a critical role helping the bacterium import metabolites (amino acids and phosphates) and enzyme factors (choline, thiamine, manganese, iron, and zinc) from the host, maintain the integrity of the outer membrane, and virulence factor secretion [93].

Additionally, the genomic searching of secretion-related genes has revealed that *C*Las lacks type III and type IV secretion systems, and some related enzymes involved in extracellular living, whereas all the genes associated with the type I secretion system and Sec pathway required for both multidrug efflux and toxin effector secretion are present in the *C*Las genome [66]. Recently, the predictions of the bioinformatic tool SecretomeP were screened with an *E. coli* alkaline phosphatase assay, with the identification of 27 non-classically secreted proteins (ncSecPs) from the *C*Las genome, where a subset of them presented higher levels of gene expression in citrus than in the psyllid vector, suppressing HR-based cell death and H_2_O_2_ accumulation in *N. benthamiana* transitory assays [96].

### 4.2. Transcriptomics

Transcriptomic profiling represents the first step to elucidate the molecular mechanisms of host–pathogen interactions. Microarrays or gene chip platforms were initially the most popular tools for transcriptomic analysis, but the emergence of RNA sequencing (RNAseq) and advancements in bioinformatics revolutionized the understanding of transcriptomic analysis. To date, short-reads high-throughput sequencing (HTS) using the Illumina platform is considered the gold standard for RNA-seq; however, long-reads sequencing platforms (Pac-Bio, and Nanopore) are emerging as more deep and accurate technologies to understand the plant–pathogen interaction transcriptome at a global level [97].

In order to find putative genetic sources of HLB tolerance, several transcriptomic studies of citrus–*C*Las pathosystems have been performed. Both susceptible and tolerant citrus species show changes mostly in gene expression related to the central metabolism (photosynthesis, glucose transportation, carbohydrate metabolism, and starch synthesis/degradation), cell wall metabolism, stress responses (abiotic and biotic), transcriptional factors, and hormone signaling [98,99,100]. On the other hand, a comparative transcriptomic analysis between HLB-susceptible and -tolerant citrus species has provided novel insights into disease tolerance. In one study, a comparative transcriptomic analysis between susceptible (sweet orange, *C. sinensis*) and tolerant (rough lemon, *C. jambhiri*) species performed at the early stage of the disease showed that the genes coding cell wall proteins could be key factors in the tolerance response [101]. In another study, the differential transcriptomic response to *C*Las infection between susceptible ‘Marsh’ grapefruit and tolerant ‘Jackson’ grapefruit identified the genes involved in pathogenesis, hormone signaling, secondary metabolism, transcription factors, and receptor-like kinases as potential key factors involved in tolerance [102]. Global transcriptome profiles of susceptible *C. sinensis* and tolerant ‘Kaffir’ lime (*C. hystrix*) showed that the cell wall metabolism, secondary metabolism, and oxidation/reduction processes may play important roles against *C*Las attack [103]. The expression analysis of *C. sinensis*, *C. sunki*, *P. trifoliata*, and contrasting hybrids representing susceptible, tolerant, and resistant HLB cultivars suggested that the upregulation of WRKY transcription factor expression and the downregulation of gibberellin synthesis are associated with *C*Las tolerance [104]. All cited transcriptomic studies have been performed, using as a reference the sweet citrus genomes of *C. sinensis* and *C. clementina*; however, the genomic source may introduce biased data for reads mapping, resulting in the loss of species-specific sequences. To overcome this limitation, a reference transcriptome for acid limes from HLB semi-tolerant ‘Mexican’ lime (*C. aurantifolia*) has been reported [105], representing an important tool for citrus–*C*Las pathosystems in cultivars mainly cultivated on the American continent. By using the *C. aurantifolia* transcriptome as a reference, a transcriptomic profile of *C. aurantifolia*–*C*Las infected plants in asymptomatic and symptomatic disease stages suggested the role of the secondary metabolism, cell wall, signaling, transcription factors, and redox reactions in HLB tolerance [106].

On the other hand, the microRNAs (miRNAs)-guided response to HLB has also been studied at the transcriptional level. microRNas are a class of small non-coding RNA sequences with 20–24 nucleotides in length, which negatively regulate gene expression by translation inhibition or transcript cleavage, that play an essential role in plant development and metabolism. Additionally, miRNAs have emerged as key regulatory molecules in response to abiotic and biotic stress, including bacterial infections [107,108]. In *C*Las-infected sweet orange (*C. sinensis*) leaves, miR399, which targets a ubiquitin-conjugating enzyme (*pho2*) involved in the degradation of phosphorus transporter proteins, is specifically upregulated. Interestingly, the treatment of plants with phosphorus oxyanion solutions induced a reduction in HLB symptoms, suggesting that a phosphorus deficiency is a key factor in HLB symptoms [109]. In another study, in roots of ‘Shanu’ tangerine (*C. reticulata*) infected with *C*Las, a total of 186 known and 71 novel miRNAs were identified, including cre-miR156a, cre-miR396b, cre-miR396g-5p, and cre-miRn70 as differentially expressed [110]. Recently, in small RNA profiling from ‘Mexican’ lime (*C. aurantifolia*) plants infected with *C*Las at asymptomatic and symptomatic stages, a total of 46 miRNAs, including 29 known miRNAs and 17 novel miRNAs, were identified. Interestingly, six miRNAs were specifically deregulated in the asymptomatic stage; meanwhile, eight miRNAs were differentially expressed in the symptomatic stage [111].

### 4.3. Proteomics

The plant–pathogen interaction triggers complex signaling events where proteome composition and protein activity are the major drivers of the infection’s final phenotype, resulting in host susceptibility or resistance. Mass spectrometry (MS)-based global proteome profiling can quantify the protein dynamics and post-translational modifications (PTMs) occurring during plant infection. Additionally, techniques such as proximity labeling, kinase-substrate profiling, and enzyme activity profiling are emerging as complementary functional tools to gain a comprehensive understanding of plant–pathogen interactions [112].

For the citrus–*C*Las interaction, proteomics can help clarify the physiological and molecular effects of disease progression, validate previous transcriptomic data, and identify early diagnostic biomarkers, short-term therapeutics, and long-term genetic resistance [113]. A proteomic approach based on two-dimensional electrophoresis (2-DE) and mass spectrometry was used to characterize the comparative proteomic changes in *C*Las-infected grapefruit plants at pre-symptomatic and symptomatic disease stages. Several downregulated proteins involved in photosynthesis and protein synthesis were linked to reduced levels of Mg, Fe, Zn, Ca, Mn, and Cu, particularly in symptomatic tissues [114]. A comparative proteomic analysis (using the iTRAQ technique) between HLB-susceptible ‘Navel’ orange (*C. sinensis*) and a moderately tolerant ‘Volkameriana’ cultivar, showed that amino acid degradation processes occurred to a larger degree in ‘Navel’ orange, whereas noticeably, four glutathione-*S*-transferases were upregulated in ‘Volkameriana’ and not in ‘Navel’ orange, highlighting the putative importance of radical ion detoxification in HLB tolerance [113]. More recently, an eight-channel iTRAQ approach was used to identify differentially expressed proteins by comparing symptomatic and healthy ‘Valencia’ orange fruits on HLB-susceptible and HLB-tolerant rootstocks. Several defense-associated proteins were downregulated in the symptomatic fruits on susceptible rootstock, including proteins involved in jasmonate signaling, jasmonate biosynthesis, vesicle trafficking, and protein hydrolysis, suggesting that jasmonate signaling and vesicle trafficking could be involved in *C*Las citrus susceptibility [115]. Interestingly, for citrus species produced on the American continent, including Mexican lime, ectopic expression α-defensin 2 and/or lysozyme induces bacterial titers, disease symptoms decrease, and photosynthesis increases in *C*Las-infected plants [116].

### 4.4. Metabolomics

During *C*Las infection, the imbalance of the host primary metabolism and failure to activate the secondary metabolism occurs in HLB-susceptible cultivars, whereas in HLB-tolerant cultivars, generally both the primary and secondary metabolism act coordinating the different defense pathways to respond against *C*Las infection [117]. Generally, susceptible varieties produce lower levels of antibacterial compounds including flavanols, flavones, and amino acid precursors of defense phenolic compounds (tyrosine, tryptophan, and phenylalanine) compared with tolerant cultivars [118]. On the other hand, HLB compromises the quality of citrus fruit and juice, causing a metallic taste, bitterness, and burning mouthfeel. By using HPLC-mass spectrometry (MS), a profile of secondary metabolites was performed in *C*Las-infected fruits of ‘Valencia’ orange (*C. sinensis*). According to Dala Paula et al. [119], significant variations were observed in sensory characteristics and secondary metabolites, with higher levels of limonin and nomilin observed in *C*Las-infected fruits. Additionally, an imbalance of volatile compounds is observed during *C*Las infection. A volatile compounds analysis in peel oil and juice from ‘Ray Ruby’ grapefruit using gas chromatography-mass spectrometry (GC-MS) showed a significantly altered volatile profile in HLB-infected fruits. *C*Las-infected fruits showed lower levels of important citrus flavor compounds including octanal, decanal, and nonanal, and also reduced levels of non-terpene compounds, terpene ketones, and other aliphatic and terpene aldehydes. The *C*Las-induced stress response was evidenced by increasing levels of ethanol, acetaldehyde, ethyl butanoate, and ethyl acetate [120]. Finally, a nuclear magnetic resonance (NMR) spectroscopy-based approach was used to identify key metabolites induced by *C*Las infection in *C. sinensis* plants. Important biomarker molecules were identified including amino acids (tyrosine, aspartate, and lysine), organic compounds (gamma butyric acid and GABA), and alcohols (4-hydroxybenzyl alcohol and p-coumaryl alcohol) [121]. HLB long incubation (6 or more months) impairs the early and opportune detection of the disease by classical molecular PCR-based methods; thus, metabolic biomarkers could be useful for this proposal on a large scale even if disease symptoms are absent.

## 5. Diagnostic Methods

Early Huanglongbing (HLB) diagnosis and the detection of infected insect vectors are essential to prevent or avoid a reduction in citrus production in orchards. The timely application of control measures is fundamental to halt the spread of this devastating disease [122,123,124,125,126,127,128,129]. Currently, one of the first steps in HLB management strategies is to identify the diseased trees by constant visual inspections in the orchards, looking with the naked eye for the characteristic symptoms of the disease. This method is difficult because a typical citrus tree has thousands of leaves, but only a few of them (usually fewer than 10) will show any symptoms. Specific parameters, such as sunlight, human training, disease severity, and field conditions, can affect the detection efficiency of this process. There are systems for the early and accurate detection of the disease, which use image analysis techniques that incorporate prior domain knowledge into hand-crafted features that an engineer adapts to describe the visual symptoms of HLB using shape, texture, color, and information on leaves and fruits [127,130]. These tools are promoted as alternatives with great potential for the permanent monitoring of crops from sowing to harvest, helping to differentiate between HLB-infected and healthy plant samples [124,126]. Samples suspected to be positive are sent to diagnostic laboratories for a secondary analysis [122,128,131,132,133,134,135]. If the disease is confirmed, control strategies are initiated, including the eradication of symptomatic trees, quarantine measures, vector control using insecticides and copper sprays [125,126,127,130,133,136,137].

Nowadays, the detection of HLB in citrus trees is difficult because HLB coexists with other citrus anomalies (i.e., other diseases, nutritional deficiencies, and pests). This complicates HLB detection because treatments focus on targeting a specific abnormality. Therefore, distinguishing between the visual subtleties of citrus anomalies is a challenge [127]. Furthermore, there is an asymptomatic period of 6–9 months after inoculation [124,126,134], where the trees appear healthy while feeding psyllids can acquire and spread the bacteria from one tree to another [138]. Foliar symptoms include yellowish spots (an uneven distribution of chlorophyll in leaves), vein plugging, starch accumulation, and leaf chlorosis, which resembles a zinc deficiency [125,127,128]. Fruit symptoms include a small size, twisted fruits, aborted seeds, bitter flavors, early fruit drop, and uneven coloration during ripening [18,125]. Once trees are very symptomatic, they usually die within five years, depending on the cultivar and their tolerance to HLB [124,125,139].

Various methodologies have been developed to detect *C*Las, including serological assays, light microscopy, electron microscopy, biological assays, DNA probes, a polymerase chain reaction (PCR), and real-time or quantitative PCR (qPCR) [127,128,129,140,141]. However, the most common methods for the molecular detection of HLB are conventional PCR with primers based on 16S rDNA targeting the three *Candidatus* Liberibacter spp. and q-PCR targeting the 16S rDNA region, as well as multiple genetic loci such as *hyvI* (*LasAI*) and *hyvII* (*LasAII*) in the *C*Las gene prophage [128,140,142]. These methodologies have several drawbacks, including the high costs of the equipment, the need for highly trained personnel, the time required to complete the process, and the difficulty of performing detection reactions under field conditions [127,128,129,141]. Despite these challenges, q-PCR has become the preferred method for detecting *Candidatus* Liberibacter species, offering the rapid and sensitive detection of these bacteria in both asymptomatic and symptomatic stages [134]. However, the results are prone to giving a false negative if the bacterial concentration of the analyzed sample is below the detection threshold of qPCR [142]. This is because they are affected by intrinsic defects that occur during certain steps of this complex technique (DNA extraction, storage, and replication), the irregular distribution of *C*Las in the leaves of infected trees, and the low concentration of *C*Las cells during the initial phase of the latency period [139,140]. For this reason, a new and accurate one-step qPCR protocol was developed to facilitate the diagnostic work of important diseases associated with the bacterial species of the *Liberibacter* genus [142]. The possibility of using it as a quick and more precise detection tool than the current ones open up.

Today, citrus producers are looking for technologies capable of detecting HLB automatically and accurately in a short time and at a low cost [127,129,130,140] and looking for alternative techniques that involve less of a sample processing time and simple equipment for rapid diagnosis [135,143]. The objective is not to replace biochemical methods, but rather to speed up diagnosis in both field and laboratory conditions.

In recent years, new protocols based on loop-mediated isothermal amplification (LAMP) and recombinase polymerase amplification (RPA) have emerged. As these techniques are a good option for rapid detection in the field, their use in the diagnosis of phytopathogenic microorganisms has increased [129]. These isothermal amplification methods were designed to detect the *C*Las species, probably because it is the most widespread species, the one most associated with enormous economic losses, and has the greatest potential for spread. Another technique for *C*Las detection combined LAMP with a lateral flow dipstick device for the visual detection of the resulting amplicons, eliminating the need for gel electrophoresis [144]. After that, an evaluation was conducted on the performance of a prototype LAMP kit for the diagnosis of HLB [128]. The results obtained showed a 100% specificity and sensitivity in the field, indicating that the prototype kit exhibits good agreement with the conventional qPCR technique. Recently, a new real-time recombinase polymerase amplification (RPA) protocol able to detect the three *Ca.* Liberibacter species associated with HLB in both plant and insect samples was implemented. Its simplicity, speed, and portability make this protocol a reliable on-site detection kit. This makes it very suitable for rapid point-of-care detection, especially useful for those disease-free countries with a high risk of the introduction of *Ca.* Liberibacter species associated with HLB [129].

Spectroscopic methods seem to be another promising alternative for the diagnosis of citrus diseases, since they can quickly measure the optical properties of samples related to their chemical composition. A study made by Cardinali et al. [136] showed that Attenuated Total Reflectance Fourier Transform Infrared (ATR-FTIR) spectroscopy enables the identification of differences between healthy, HLB-symptomatic, and HLB-asymptomatic leaves of sweet orange trees, with a success rate of 94%. This method has the potential to be used as a large-scale citrus disease diagnostic tool, aimed at constructing infestation maps, and therefore, providing a more efficient HLB management strategy. Furthermore, a methodology was developed based on laser-induced breakdown spectroscopy (LIBS) for rapid and efficient discrimination between healthy and HLB-affected citrus plants by directly analyzing portions of the phloem, the affected tissue [139]. This same LIBS was used by Killiny et al. [125] to reveal key biochemical differences between *C*Las-infected and uninfected psyllids, demonstrating that it allows the rapid and reliable detection of the *C*Las pathogen in *D. citri*. Also, HLB detection protocols based on LIBS combined with chemometric strategies have been developed to successfully predict the condition of *C*Las-infected orchard plants [138]. Other methodologies are based on the mass spectrometry imaging technique that allows the detection of increased concentrations of metabolites associated with the disease (quinic acid, phenylalanine, nobiletin, and sucrose) [140] and the spectroscopy of fluorescence imaging to detect the main diseases that affect citrus and differentiate between samples with HLB or zinc deficiency stress [143]. Neves et al. [135] developed a multi-class classifier that discriminates citrus diseases using fluorescence imaging spectroscopy, proving to be a viable approach to help early diagnosis in the field and an effective method for diagnosing diseases in citrus.

## 6. Disease Management Methods

Since the arrival of HLB in Latin America in 2004, disease management has focused on preventing its spread, especially in Brazil and Mexico, the main citrus-producing countries in the region.

### 6.1. Government Policies

In Brazil, in 2005, a HLB Suppression Program was implemented in São Paulo, Brazil, led by the Ministry of Agriculture, Livestock, and Supply (MAPA), with support from the Fundo de Defesa da Citricultura (FUNDECITRUS). The main objective was to eliminate symptomatic trees and raise growers’ awareness of the importance of removing inoculum sources, regulated under the legislative control of the first Normative Instruction (IN10-2005). The management was based on a triple system, involving the use of certified nursery plants, monitoring, and control of *D. citri*, and the detection and eradication of symptomatic HLB trees in commercial orchards [2,12,145]. The initial regulation was replaced by IN32-2006, transferring the responsibility for detecting and removing symptomatic trees to growers, while MAPA and FUNDECITRUS only carried out control inspections. Later, these regulations were replaced by IN53-2008, which incorporated the State Agency for Plant Health Defense (OEDSV) into inspections and established a strict strategy: removing only symptomatic trees if the infection rate was below 28% or eradicating the entire orchard if it exceeded this threshold [12,145]. These regulations remained in effect until 2021, when MAPA issued Decree SDA317-2021 and established the National Program for HLB Prevention and Control (PNCHLB). Based on this new regulation, the São Paulo Department of Agriculture and Supply (SAA) published Resolution SAA88-2021, which incorporated Integrated Pest Management (IPM) and required the removal of symptomatic trees up to eight years old, along with mandatory monitoring and fumigation records for five years by growers [146,147].

In Mexico, after the first report of HLB disease in the south of the country in 2009, the government implemented a “Phytosanitary campaign against HLB” and the publication of the Emergency Mexican Official Standard (NOM-EM-047-FITO-2009) [148] as a strategy to prevent the spread of it to the north of the country. But by 2010, the disease was dispersed throughout the Pacific and the northwest states of Mexico. To restrict the spread of HLB and its establishment in other citrus-growing regions, it was prohibited to transport nursery plants from these areas to states that are free from HLB. Additionally, the monitoring, control of *D. citri*, and detection and eradication of symptomatic HLB trees in commercial orchards were implemented based on the experiences of Brazil and other countries. Producers were also advised to manage the vector in commercial orchards by alternating biological and chemical applications and utilizing certified nursery plants in replants and new plantations [149,150,151]. Before the detection of HLB in Brazil, citrus producers implemented control measures recommended for citrus canker and citrus variegated chlorosis, including exclusion, eradication, and protective management practices. Based on the experiences and knowledge gained from these diseases, prevention recommendations for HLB were easily applied [12]. In Mexico, the federal government took phytosanitary actions proactively to mitigate the risk of HLB introduction and spread, starting in 2008, a year before the first detection, through the national priority phytosanitary campaign against HLB, due to its proximity to countries where this disease had al-ready been detected [152]. From 2012 to 2018, SENASICA established monitoring and management zones through the HLB and *D. citri* Regional Areas for the Control (ARCOs). This program was designed as a model to demarcate monitoring and control areas. In 2019, these zones were replaced by the Phytosanitary Epidemiological Management Areas (AMEFIs), and the phytosanitary campaign was modified into the “Phytosanitary Protection Campaign for Citrus Pests” [153].

Plenty of efforts have been made to struggle with the disease, applying different management strategies such as chemical control, biological control, and cultural control to mitigate the significant economic losses caused by HLB. Currently, there is no available method for successful HLB management; therefore, the disease is treated preventively, and management techniques can target both the vector insect and the causal agent [12].

### 6.2. Chemical Management

The widely used strategy in Latin America focuses on reducing the populations of the vector *D. citri* to mitigate the risk of HLB spread by employing chemical control. The most frequently used insecticides in Brazilian citriculture are systemic insecticides such as thiamethoxam, chlorantraniliprole, imidacloprid, and dimethoate. These insecticides interfere with the developmental processes of *D. citri* in a way that affects the transmission efficiency of *C*Las, reducing it by up to 59% [154,155]. Another group of used insecticides are contact insecticides like bifenthrin, malathion, chlorpyrifos, paraffin oil, and mineral oils. In Mexico, they have been used since 2007 for *D. citri* control due to their low cost. However, producers have reported reduced effectiveness starting in 2011 [156,157,158]. Adult and fourth-stage nymphs of *D. citri* exhibit high resistance to malathion and chlorpyrifos in some citrus-producing areas in Mexico. The biological effectiveness of chlorpyrifos in combination with paraffin oil can be improved, achieving up to 71% nymph mortality. However, it is reported that insecticides belonging to the toxicological groups of mineral oils, heterocyclic S-ethyl organophosphates, and spinosins exert greater absolute and relative selection pressure in Mexico [156,158]. To maximize the efficiency of insecticide use, the effectiveness of foliar applications has been evaluated to reduce environmental impacts and production costs. For example, the residual efficiency of insecticides such as dimethoate, imidacloprid, and bifenthrin have been documented up to 14 days after application, with a 50% psyllid mortality rate. Optimal coverage can be achieved using between 25 mL and 40 mL of spray/m^3^, in contrast to the standard 70 mL/m^3^ used by citrus growers in Brazil. This demonstrates the potential to reduce water usage, insecticide rates per hectare by up to 64%, and production costs by 40% [157]. Another factor that influences the effectiveness is the tree stage, as mentioned by de Carli et al. [159], where the effectiveness of insecticides against adult *D. citri* in the vegetative growth stages of ‘Valencia’ orange trees is lower, unlike mature leaves, especially for imidacloprid and bifenthrin. Therefore, they suggest frequent applications during the young shoot stages. Other optimization methods tested in Latin America, mainly in Brazil, include the nanoencapsulation of thiamethoxam in polymeric micelles, showing that the nanoformulation is effective in controlling *D. citri* at doses approximately two times lower than standard formulations [160]. Additionally, the combination of insecticides used with trap plants, such as the curry tree (*Bergera koenigii* L.), may be a strategy with the potential to increase the effectiveness and reduce the use of systemic insecticides in citriculture. The curry tree is more attractive to *D. citri* than citrus trees, and it is immune to HLB-causing bacteria. The technique is based on creating an attraction and death barrier, and insecticides control adults for long periods on this tree, up to 42 days after applications [154]. In Mexico, silver nanoparticles from Argovit™ (AgNP) have been tested in ‘Mexican’ lime orchard trees to directly target *C*Las bacteria through foliar spraying and trunk injection. Both methods achieve reductions in bacterial titers between 80% and 90%. Additionally, they reduce starch accumulation in phloem vessels, unlike β-lactam antibiotics, with greater potency of AgNP [161].

### 6.3. Sustainable Management

The natural enemies of any vector, such as parasitoids, predators, and pathogens, are the primary factors of natural mortality control in ecosystems and are key components for integrated pest management. The psyllid *D. citri* can be controlled by biological control agents such as parasitoids *Tamarixia radiata* and *Diaphorencyrtus aligarhensis*, among others. Some entomopathogens like *Hirsutella citriformis* can function as biopesticides [162]. Nymphs of *D. citri* are naturally parasitized by *T. radiata*, and the presence of this parasitoid has been reported in citrus orchards in Brazil, Ecuador, and Mexico [163,164,165]. In Brazil, the successful parasitism of *D. citri* nymphs by *T. radiata* can reach up to 90%, but this depends on environmental conditions, with higher levels of parasitism observed in the summer [163]. In other citrus regions, *T. radiata* can cause parasitism rates of around 60%, as reported in Mexico [165]. The parasitism of *D. citri* by the wasp *Diaphorencyrtus* sp. has also been documented in Colombia, Mexico, and Ecuador, although the successful establishment of this parasitoid has not been entirely achieved [51,165,166,167]. Mites of the *Phytoseiidae* family also have the ability to prey on *D. citri* eggs. For instance, the mite *Amblyseius herbicolus* has been experimentally observed to parasitize up to five eggs per day. *A. herbicolus* is present in citrus orchards, where the densities of adult and nymphal *D. citri* are up to 85% lower than in trees without the mite [168,169]. Other natural enemies of *D. citri* include *Chrysoperla* sp. and ladybugs such as *Cycloneda sanguínea* and *Olla v-nigrum* [165]. Some entomopathogens like the fungus *H. citriformis* have high virulence against both adult and nymphal *D. citri* strains from citrus regions in Mexico, which achieved mortality rates of up to 88% and 82% for adults and nymphs of the psyllid, respectively [170,171]. Strains of the fungi *Metarhizium anisopliae*, *Cordyceps bassiana*, and *Isaria fumosorosea* have also proven to be effective against nymphs and adults of *D. citri* under field conditions [172]. In Brazil, the translocation of endophytic strains of *Bacillus thuringiensis* in citrus seedlings from roots to shoots has been tested. The systemic pathogenicity for third-stage nymphs of *D. citri* reached up to 90% mortality. *B. thuringiensis* has the potential to be used as a bioinsecticide [173]. The exogenous application of some oils, plant extracts, and secondary metabolites has been effective against *D. citri*. In Mexico, applications of different oils (paraffin, cooking, citrus, and Jatropha seed oil), neem, garlic, and onion extract on young ‘Mexican’ lime shoots have shown repellent effects, reducing *D. citri* infestation. In vitro, other essential oils from plants, such as anethole, verbenone, 4-ethyl-4-methyl-1-hexene, 4-allylanisole, and trans-tagetone, as well as extracts from *Foeniculum vulgare* and *Tagetes* sp., have been shown to be toxic or repellent to *D. citri* adults and nymphs. The greatest biocontrol potential is attributed to Tagetes sp. oil, with up to 92% efficacy at a dose of 40 mg mL-1 [174,175]. Secondary metabolites (F4A) from *Pseudomonas aeruginosa* have the potential to be used as bioinsecticides. In greenhouse tests, when applied as a spray on HLB-infected citrus trees, they reduced the bacterial titer and induced the overexpression of defense genes [176]. Another biological control alternative with the ability to reduce the *C*Las bacterial titer and induce defense genes is the spray application of plant brassinosteroids, with epibrassinolide showing promising results [177].

### 6.4. Cultural Management

Agronomic practices, such as improvements in irrigation, nutrition, pruning, the tree density, and weed control, can contribute to the management of HLB and are recommended practices for increasing citrus productivity. Nutritional deficiencies caused by poor water and nutrient absorption because of root system damage caused by HLB lead producers to adopt nutrient application practices [178,179]. Supplying micronutrients (Zn, Mn, and Cu) individually helps mitigate the harmful effects of HLB on the starch metabolism, but it does not reduce the *C*Las titer [178]. On the other hand, supplying calcium (Ca), potassium (K), and silicon (Si) can induce a certain degree of plant defense. The effects of foliar and soil nutrient applications (Ca, K, and Si) reduce *D. citri* populations, indicating that these mineral nutrients can enhance plant resistance and be applied in management programs [180]. In Brazil, through a progressive treatment with foliar fertilization and resistance inducers, temporary nutritional standards for plants damaged by HLB were restored. However, nutritional stability was not maintained over long periods of time [181]. In Mexico, it has been reported that chemical fertilization (100 N—22 P_2_O_5_—195 K_2_O—30 MgO) and combined fertilization (chemical and compost) applied to soil, plus foliar fertilization with zinc, sulfate, iron, copper, manganese, and borax increased flowering and fruiting [182]. The combination of slow-release fertilizers and soluble fertilizers is more efficient than conventional slow-release fertilizer-based fertilization. The production is similar, and a considerably lower number of nutrients are used. However, the treatment effects were not consistent throughout the year in evaluations conducted in Puerto Rico; nonetheless, it improved the crop’s profitability [183]. On the other hand, Bassanezi et al. [179] mentioned that not all nutritional programs and exogenous hormones (auxin) applications prove to be effective against HLB. Implementing these methods may favor the accumulation of inoculum within a region and facilitate HLB spread by the vector. Therefore, rigorous vector control using insecticides and the eradication of diseased plants continue to be the best long-term HLB control methods. There are alternatives to cultural management, such as increasing the tree density per hectare. In Brazil, it has been proven that high tree densities (from 220 to 714 trees/ha) reduce HLB incidence, in addition to *D. citri* control and the removal of symptomatic trees [184]. The use of plastic mulch in black, white, and aluminum has a positive effect on reducing HLB incidence and severity by up to 40% over a 13-month period. It also reduces the number of *D. citri* adults and increases the fruit yield. Therefore, this production system is a viable alternative for growers to coexist with HLB [185]. Other practices have been documented, such as the application of kaolin spray to deter *D. citri*; this technique can reduce psyllid populations by up to 80%. The application of kaolin (2%) at intervals of 7 to 14 days can be a sustainable tool to reduce psyllid infestation and spread [157].

### 6.5. Emerging Biotechnology Approaches

The previously described management strategies have not shown completely effective results; therefore, the struggle against HLB disease continues. An alternative that could prove entirely effective, environmentally friendly, and economically viable for producers is genetic control through disease resistance. However, to date, no sources of resistance have been detected in citrus germplasm. Multiple ongoing lines of research have the potential to develop alternatives, with emerging biotechnologies playing a prominent role in finding solutions to the disease [50,162,186]. In Latin America, genetic engineering has been explored through the transformation of exogenous genes, such as those encoding antimicrobial peptides and bacterial toxins, as a possible strategy. The goal is to generate transgenic citrus crops with resistance or tolerance to HLB and its vector. In Brazil, the genetic transformation of sweet orange plants expressing the *attA* gene, which encodes the antibacterial peptide Atacina A, has been reported. These transgenic plants did not reduce the *C*Las titer, but showed the increased development of new shoots. The authors associated this response with tolerance to the disease and highlighted the potential of the *attA* gene for HLB management [187,188]. Other transgenic sweet orange plants carrying the AMP sarcotoxin IA (*stx IA*) transgene, an antimicrobial peptide secreted by the larvae of the flesh fly *Sarcophaga peregrina*, did not exhibit resistance responses to *C*Las infection, but some lines showed fewer HLB symptoms [189]. The major challenge is to identify potential genes for generating transgenic or edited citrus plants. Genes targeting the *D. citri* vector have also been tested. Some transgenic ‘Hamlin’ and ‘Valencia’ orange plants carrying the *cry11A* gene from *Bacillus thuringiensis*, with bioinsecticidal activity against *D. citri* nymphs, caused mortality levels ranging from 22% to 43% in nymphs. The researchers mention that they have identified other *cry* genes with greater effectiveness, which will be used in developing new transgenic lines, emphasizing the potential of these genes for managing both *D. citri* and HLB [190].

Another emerging biotechnology is RNA interference (RNAi), taking advantage of a natural mechanism of gene regulation and antiviral defense system in eukaryotic cells. This mechanism allows the design of double-stranded RNA (dsRNA) for use as pesticides by its exogenous application. Recent research has demonstrated the feasibility of RNAi-based strategies to control insect pests and pathogens. This technology is being tested against *D. citri* during natural feeding on dsRNA-treated citrus flushes [191]. A study reported by Andrade and Hunter [192] showed that feeding psyllids with shoots treated with dsRNA targeting the *D. citri* arginine kinase gene (*dsAK*) interrupted transcription and caused psyllid mortality, demonstrating a robust RNAi response. Additionally, directing interference to the genes cathepsin D, chitin synthase, and the apoptosis inhibitor of adult and nymph *D. citri* through artificial diets with dsRNA caused mortality in both adults and nymphs [193]. Another study demonstrated that genes encoding effectors of *D. citri*, overexpressed in the head and manipulated by *C*Las, can be targeted for RNAi-induced silencing. The administration of dsRNA through feeding eliminated *D. citri* effectors and interfered with the feeding capacity and survival of the psyllid [194]. In Mexico, a method based on small interfering RNA (siRNA) generated through Escherichia coli has been evaluated. These siRNAs were tested against genes such as abnormal wing disc (*AWD*), superoxide dismutase 1 (*SOD1*), and wingless (*WNT*), which play key roles in the development and maturity of the psyllid. The siRNAs induced the silencing of all three genes, reducing gene expression and causing psyllid mortality, with greater effectiveness against the *WNT* gene [195]. These new approaches highlight the potential of a functional RNAi system that could be exploited for the successful management of HLB.

## 7. Conclusions

Huanglongbing is an epidemiological issue in Latin America, influenced by the diversity of climatic conditions, the introduction of *C*Las to these countries, and the genetic diversity worldwide. This makes it difficult to control effectively. Information about the tripartite interaction among the bacteria, host, and vector has been provided by the application of omics approaches, such as genomics, proteomics, and metabolomics. Early detection remains a challenge, so rapid and accurate methods have been evaluated to detect the bacteria in early stages. Image analysis techniques that incorporate prior domain knowledge into hand-crafted features that an engineer adapts to describe the visual symptoms of HLB have been developed for the early and accurate detection of HLB disease. However, these techniques are difficult because HLB coexists with other citrus anomalies (i.e., diseases, nutritional deficiencies, and pests). This makes HLB detection more complex because treatments focus on targeting a specific abnormality. Therefore, distinguishing between the visual subtleties of citrus anomalies is a challenge. Since the arrival of HLB in Latin America in 2004, disease management has focused on preventing its spread, especially in Brazil and Mexico, the main citrus-producing countries in the region. In Brazil, the management was based on a triple system, involving the use of certified nursery plants, the monitoring and control of *D. citri*, and the detection and eradication of symptomatic HLB trees in commercial orchards. Various efforts have been made to deal with the disease, applying different management strategies such as chemical control, biological control, and cultural control to mitigate the significant economic losses caused by HLB. Currently, there is no available method for successful HLB management; therefore, the disease is treated preventively, and management techniques can target both the insect vector and the causal agent. RNA interference (RNAi) and small interfering RNA (siRNA) technologies offer promising options for HLB management. However, significant research and development efforts are needed to address the economic and technical challenges associated with their implementation. It is imperative to strengthen collaboration between countries and invest in research to deal with this ongoing threat to citrus farming in the Latin American region.

## Figures and Tables

**Figure 1 biology-14-00335-f001:**
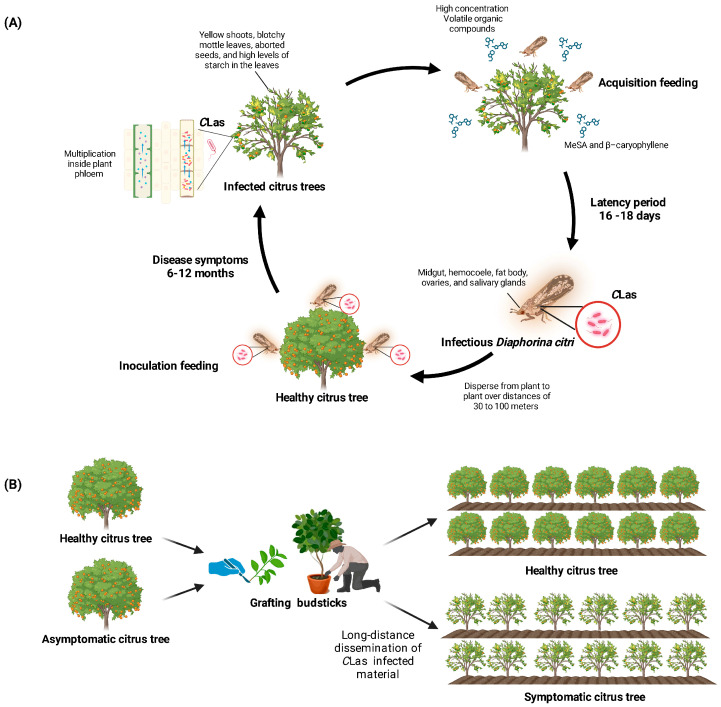
*Candidatus Liberibacter asiaticus* infection cycle. (**A**) Acquisition by *Diaphorina citri*: The healthy insect acquires *C*Las by feeding on infected citrus plants, which are more attractive due to the high concentration of volatile organic compounds such as MeSA and β-caryophyllene. For *D. citri* to become infectious, it must undergo a latency period of 16–18 days. After this period, it can spread the disease from an infected citrus plant to a healthy one, with the first symptoms appearing 6 to 12 months after inoculation. *C*Las multiplies within the phloem [2,32,33,34,35,36,37,38,39]. (**B**) Long-distance spread: *C*Las can be transmitted over long distances through grafting from asymptomatic infected citrus trees and the movement of infected plant material [27]. This figure was created using BioRender. “https://BioRender.com (accessed on 19 March 2025)”.

**Figure 2 biology-14-00335-f002:**
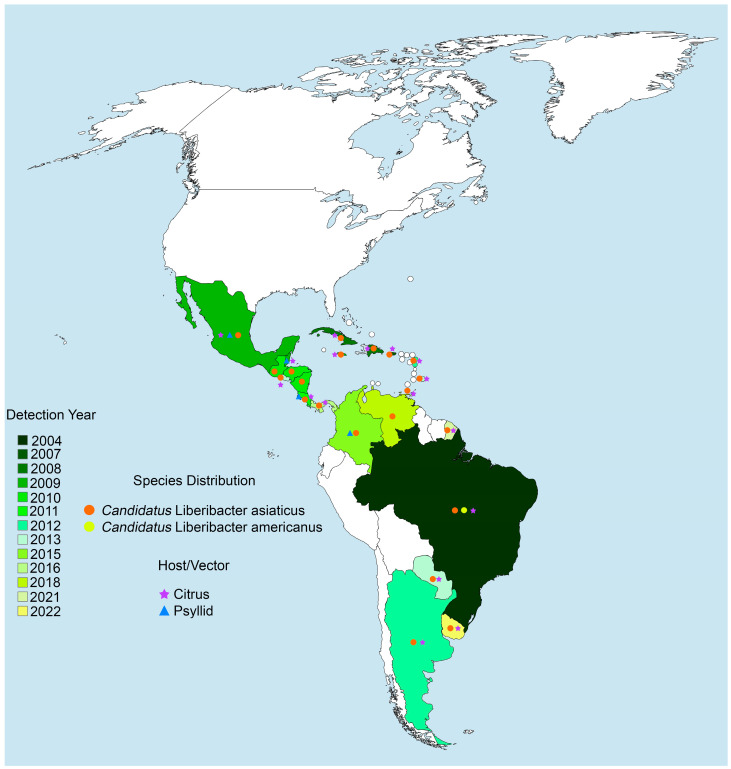
Distribution of Huanglongbing (HLB) and detection of species of *Candidatus* Liberibacter in citrus and insect vector in Latin America from 2004 to 2022, EPPO Global Database information. This figure was created using MapChart “https://www.mapchart.net (accessed on 19 April 2024)”.

**Figure 3 biology-14-00335-f003:**
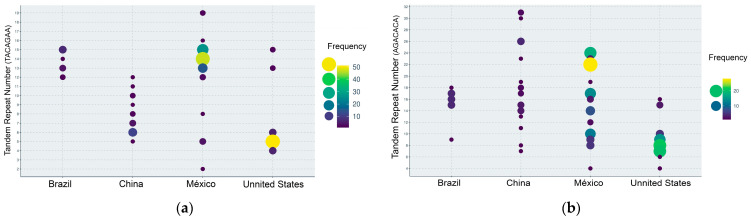
Genetic diversity of *Candidatus* Liberibacter asiaticus (*C*Las) strains from Brazil, China, Mexico, and the United States by double-locus analyses. (**a**) Distribution of TRN frequencies using repeat unit TACAGAA; (**b**) Distribution of TRN frequencies using repeat unit AGACACA. This graphic was performed using R Software (v4.3.2; R Core Team) with data published by [68,69,70,71].

**Figure 4 biology-14-00335-f004:**
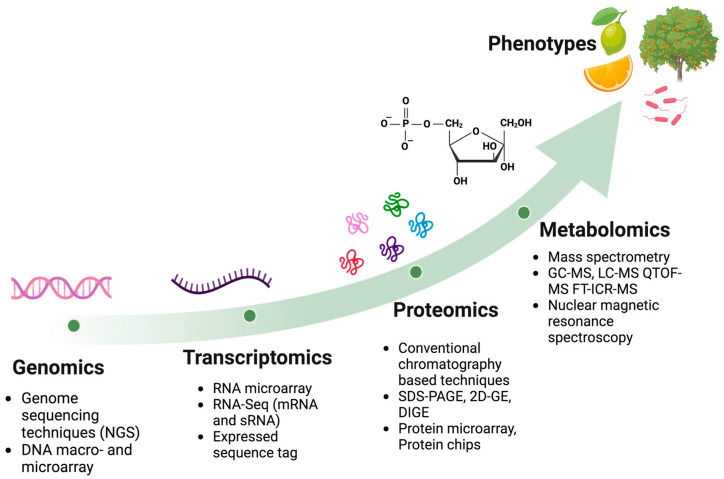
Scheme of “Omics” approaches. Genomics, transcriptomics, proteomics, and metabolomics are the compressive analysis targeting genome, transcriptome, proteome, and metabolome, respectively, to elucidate the molecular regulation of different citrus phenotypes. The main instrumental tools used for each “Omic” approach are shown. This figure was created using BioRender. “https://BioRender.com (accessed on 8 November 2023)”.

**Table 1 biology-14-00335-t001:** *Candidatus* Liberibacter species associated with HLB disease.

Species	Subsp.	Abbr.	Characteristics	Distribution	Reference
*Candidatus* Liberibacter africanus	capenis	*C*LafC	Heat-sensitive (22–24 °C)	Africa	[2,16,17,18]
	clausenae	*C*LafCl			
	zanthoxyli	*C*LafZ			
	vepridis	*C*LafV			
	tecleae	*C*LafT			
*Candidatus* Liberibacter asiaticus		*C*Las	Heat-tolerant (≤35 °C)	Asia, Oceania, America, and Africa (Ethiopia and Kenya)	[2,15,19,20]
*Candidatus* Liberibacter americanus		*C*Lam	Heat-sensitive (≤32 °C)	Brazil	[15]

**Table 2 biology-14-00335-t002:** *Candidatus* Phytoplasma is associated with symptoms of HLB disease.

Candidatus Phytoplasma	Distribution	Reference
*Candidatus* Phytoplasma phoenicium	Brazil	[21]
*Candidatus* Phytoplasma asteris	China Mexico	[22,23]
*Candidatus* Phytoplasma aurantifolia	China	[24]
*Candidatus* Phytoplasma pruni	Brazil	[25]
*Candidatus* Phytoplasma ulmi	Chile	[26]
*Candidatus* Phytoplasma hispanicum	Chile	[26]

**Table 3 biology-14-00335-t003:** *Liberibacter* genomes described in America available in the GenBank database (version 249) (https://www.ncbi.nlm.nih.gov/genbank/release/249/ accessed on 15 January 2025).

Species	Strain	GenBank Accession Number	Geographic Area	Size (Mb)	No. of Genes
*C*Las ^1^	SGCA16	VTLZ01	California, USA	1.21	1102
	SGCA5	LMTO01		1.20	1112
	AHCA1	CP029348.1		1.23	1110
	AHCA17	VNFL01		1.21	1103
	A-SBCA19	JADBIB01		1.19	1126
	HHCA	JMIL02		1.15	1212
	HHCA16	VTLY01		1.21	1121
	DUR1TX1	VTLT01	Texas, USA	1.21	1098
	DUR2TX1	VTLS01		1.21	1161
	GFR3TX3	VTLR01		1.21	1109
	LBR19TX2	VTMA01		1.20	1085
	LBR23TX5	VTMB01		1.20	1087
	TX2351	MTIM01		1.25	1191
	CRCFL16	VTLW01	Florida, USA	1.21	1187
	FL17	JWHA01		1.23	1116
	MFL16	VTLX01		1.20	1134
	JRPAMB1	CP040636.1		1.24	1119
	Psy62	CP001677.5		1.23	1114
	CoFLP	CP054558.1	La Guajira, Colombia	1.23	1114
	9PA	JABDRZ01	Brazil	1.23	1113
	Mex8	VTLU01	Baja California, Mexico	1.24	1141
	BCSMX	JAOPHS01	Baja California Sur, Mexico	1.23	1116
	YTMX	JAOPHR01	Yucatan, Mexico	1.23	1111
*C*Lso ^2^	RSTM	LLVZ01	California, USA	1.29	1210
	R1	JNVH01		1.20	1143
	CLso-ZC1	CP002371.1	Texas, USA	1.26	1169
	HenneA	JQIG01		1.21	1146
*C*Lam ^3^	PW_SP	AOFG01	Sao Paulo, Brazil	1.17	1018
	Sao Paulo	CP006604.1		1.19	1045
Lcr ^4^	BT-0	CP010522.1	Puerto Rico	1.52	1389
	BT-1	CP003789.1		1.50	1388

^1^ *Candidatus* Liberibacter asiaticus (*C*Las), ^2^ *Candidatus* Liberibacter solanacearum (*C*Lso), ^3^ *Candidatus* Liberibacter americanus (*C*Lam) ^4^
*Liberibacter crescens* (Lcr).

## Data Availability

No new data were created.

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
