# Peer review of "Huanglongbing as a Persistent Threat to Citriculture in Latin America"

_biology, 2025, doi:10.3390/biology14040335_

Round 1

Reviewer 1 Report

Comments and Suggestions for Authors

This review paper is a decent summary of the HLB situation and HLB-related research, with an emphasis on Latin America, particularly Brazil and Mexico. It could have been improved with additional information from more Latin American countries, but it is understood that Brazil and Mexico are the largest citrus producers in Latin America and so this emphasis is understandable. The authors might make a statement regarding this towards the beginning of the submission. This reviewer believes that the citations in the text could be improved, either with additional citations or in the placement of the citations. The references themselves seem adequate. General comments on the English are in the section below and, in this reviewer's opinion, should mostly be the responsibility of a copy editor. However, some comments for the authors that will improve the submission are:

32 – 35 and 73 – 78: "restricted to developed countries": implies all Latin American countries are undeveloped, which this reviewer does not think is the case. This is written without knowing if there are actually any criteria regarding use of this term.

46: substitute "immense" for "whopping"

89: substitute "and" for "y"

93: substitute "budsticks" for "rods"

101: "bacterium" for "bacteria"

124 – 125: "has also spread" in place of "is also spread"

142 – 146: "movement" in place of "displacement"

176 – 177: This reviewer is not sure that CLam was the first reported C Liberibacter. For instance, Bassanezi et al 2020 https://doi.org/10.1007/s40858-020-00343-y states: "HLB was first reported in Brazil in 2004 (Coletta-Filho et al. 2004; Teixeira et al. 2005a). Initially, the disease was limited to the municipalities located in the center of Sao Paulo State. ‘Candidatus Liberibacter asiaticus’, the bacterium species described as the associated agent of HLB in Asian countries, was rarely detected in these early cases. Less than 2% of leaf samples that showed typical symptoms of HLB, namely, diffuse and asymmetrical chlorosis (known as leaf mottling), contained 'Ca. Liberibacter asiaticus’. The common agent that was present in over 98% of the ‘Ca. Liberibacter’-positive samples was a new organism, since characterized as a new species of ‘Ca. Liberibacter’ and named ‘Ca. L. americanus’ (Teixeira et al. 2005a, b)." Suggest the authors double check on this aspect.

243: "regions containing both sweet and sour citrus samples": please clarify that this refers to sweet orange and some other type of citrus...does sour refer to sour orange, lemon, grapefruit? Latter 2 are grown in southern Mexico, sour orange would be extremely unusual there.

244 – 251: the 2 described genogroups appear to coincide with temeperate (1) vs tropical (2) environments. Is this related to the statement in 244 – 245.

284 – 286: probably should have reference(s) here

298: "comprehensive" in place of "compressive"

302 – 303: nice graphic

318 – 321: needs rewriting

343 – 345: previously, only CLas, CLaf, and CLam were mentioned. Newly named strains should be identified better, either earlier or perhaps more appropriately in this section.

347 – 354: need references

416: "Marsh" grapefruit (not "March")

540 – 549: First, it is stated that "the most common methods...are conventional PCR" and later "...q-PCR has become the preferred detection method...". These statements seem contradictory. It appears to this reviewer that qPCR is the most common and preferred detection method, although of course this reviewer does not know on a global scale what all diagnostic labs are doing.

604: This reviewer believes this section should mention the coordination of grower efforts in combating HLB/ACP. These are ARCOs in Mexico but this reviewer doe not remember how they are called in Brazil. This might go in section 6.1 or 6.2. This reviewer supposes that these coordinated efforts are still in place. If not, mention should be made of their establishment and why they were discontinued.

650: "To maximize the use of insecticides". The authors probably mean "To maximize the efficiency of insecticide use".

842 – 1263: References not reviewed.

This reviewer looks forward to clipping the final, published version.

Comments on the Quality of English Language

Although generally well written and understandable, there are numerous small corrections that should be made to the English. A few comments related to the English are made above for the authors's attention. The remainder would, this reviewer believes, be made by a copy editor rather than the authors. 

Reviewer 2 Report

Comments and Suggestions for Authors

pls. see attached comments

Reviewer 3 Report

Comments and Suggestions for Authors

Dear Authors,

This review manuscript was well organized and it included a lot of detailed knowledge about Huanglongbing. English was used very well. However, it needs a small revision based on the names of the cultivars and some references. They were marked on the manuscript uploaded. After this revision, it will be suitable for publication. There is no need for me to check the manuscript again. 

All best, 
